# Unveiling the impact of community knowledge in malaria programmes: A scoping review protocol

**Faizul Akmal Abdul Rahim** *, **Mohd Hatta Abdul Mutalip, Ahmad Mohiddin Mohd Ngesom, Mohd Amierul Fikri Mahmud, Norzawati Yoep**

Centre for Communicable Diseases Research, Institutes for Public Health, National Institutes of Health, Ministry of Health, Shah Alam, Malaysia

* faizul.fabregas@gmail.com

## Abstract

### Background

Despite significant reductions in recent malaria cases and deaths globally, the persistence of this health concern necessitates a shift from traditional top-down approaches. Consequently, malaria control initiatives increasingly focus on empowering local communities through community-centred strategies. Therefore, this scoping review protocol systematically explores diverse community knowledge approaches adopted in malaria programmes worldwide and their associated outcomes.

### Methods

Adhering rigorously to the Preferred Reporting Items for Systematic Reviews and Meta-Analyses Extension for Scoping Reviews (PRISMA-ScR) guidelines, a comprehensive scoping review protocol was developed. Collaborating with a research librarian, a systematic search strategy targeted peer-reviewed literature from databases such as PubMed, Embase, Scopus, and Web of Science, complemented by a thorough grey literature search. Titles and abstracts will be screened, followed by extracting bibliographic details and outcome information using a standardized framework. Subsequently, the results will be systematically summarized and presented in a structured tabular format (S1 Checklist).

### Discussion

This scoping review promises an in-depth understanding of current research regarding the impact of community knowledge in malaria programmes. The identification of knowledge gaps and intervention needs serves as a valuable resource for malaria-affected countries. The profound implications of community knowledge underscore its pivotal role in enhancing the effectiveness of prevention, control, and elimination efforts. Insights from this review will assist policymakers, empowering implementers and community leaders in designing effective interventions. This concerted effort aims to adeptly leverage

**Data Availability Statement:** No datasets were generated or analysed during the current study as it was a protocol for a scoping review. All relevant

data from this study will be made available upon study completion.

**Funding:** The author(s) received no specific funding for this work.

**Competing interests:** The authors have declared that no competing interests exist.

community knowledge, thereby propelling progress toward the achievement of malaria elimination goals.

## Introduction

### Background

Malaria, a vector-borne disease caused by Plasmodium parasites, remains a significant global health challenge. In 2020 alone, it accounted for an estimated 241 million cases and 627,000 deaths across 85 endemic countries [1]. Over the two decades from 2000 to 2020, the world grappled with a staggering burden of 1.7 billion cases and 10.6 million deaths. Notably, the WHO African Region witnessed 82% of these cases and 95% of deaths, while the WHO South-East Asia Region contributed 10% of cases and 2% of deaths [1].

Countries affected by malaria have implemented diverse strategies for prevention and control, ranging from the use of insecticide-treated nets (ITNs), long-lasting insecticidal nets (LLINs), indoor residual spraying (IRS), rapid treatment of diagnosed cases, and administration of antimalarial drugs [2]. While these strategies have achieved significant reductions in malaria cases and deaths, there is a pressing need for a more comprehensive approach to further enhance outcomes [2–4]. In response to this persistent challenge, healthcare professionals, researchers, and policymakers increasingly recognise the pivotal role that communities play in the battle against malaria [5, 6]. Recent years have seen a growing acknowledgment of the indispensable role that community knowledge plays in this multifaceted campaign. The global approach to combat this infectious disease has shifted from conventional top-down methods to more inclusive, community-centred strategies, emphasizing the inherent value of community knowledge in malaria prevention, control, and elimination [7, 8].

Community knowledge of malaria refers to the community's understanding and awareness of the malaria disease. This includes a comprehensive knowledge of the transmission, symptoms, transmission, malaria risk activities, and preventive measures [9]. Understanding community knowledge is essential for designing effective malaria control and elimination strategies, tailored to the specific needs and beliefs of the communities they serve. It also helps dispel misconceptions and overcome barriers that may hinder the adoption of recommended malaria prevention and treatment practices [9, 10]. Harnessing the power of community knowledge enables the formulation of interventions and strategies that are not only more efficacious but also sustainable. As nations strive for malaria-free status, insights from this investigation will inform the development of tailored communication and education initiatives, empowering communities to actively engage in malaria surveillance, reporting, and control [11]. Community education programmes have emerged as vital instruments in raising awareness and fostering active participation in malaria control efforts, essential components on the journey toward malaria elimination [9, 12, 13]. These educational initiatives have the potential to empower community members by deepening their understanding of the disease, its prevention, and treatment. The use of interactive educational curricula, integrating elements like theatre, music, and engaging discussions, has proven to be a highly effective strategy for engaging community members and enhancing their understanding and awareness of malaria [14, 15].

Within the sphere of malaria control, participatory learning techniques have been well-documented for their efficacy in instigating behavioural changes [15]. Education and optimise the impact of community-based malaria control programmes, enhancing the knowledge and behaviours of community members becomes imperative [12, 13]. This heightened community

knowledge about malaria has translated into a higher utilisation rate of bed nets. Additionally, educational initiatives have revealed a significant correlation with individual-level adoption of ITNs [16]. Studies in Myanmar [17] and Tanzania [18] have indicated that the use of ITNs witnessed an upsurge when individuals were exposed to health promotion activities concerning ITNs. Remarkably, these educational programmes have also proved advantageous for healthcare professionals by augmenting their understanding of surveillance principles, case management, and vector control [17, 18].

## Rationale

Communities are integral to the success of malaria programmes, playing a pivotal role in shaping outcomes. Understanding their knowledge is paramount for tailoring interventions to specific cultural, social, and environmental contexts. This reservoir of community knowledge not only directly influences preventive behaviours, treatment-seeking patterns, and overall engagement in malaria control efforts but also acts as a barometer for the community's response to interventions [19, 20]. Through in-depth analysis, gaps and misconceptions can be identified, crucial for designing targeted educational campaigns. Addressing specific barriers to preventive measures, such as bed net usage, indoor residual spraying, and antimalarial drug adherence, becomes more effective with insights from community knowledge. Early detection of malaria cases is facilitated by the community's understanding of symptoms and transmission dynamics, thereby minimizing disease severity and spread [21]. Additionally, informed communities actively participate in vector control measures, such as environmental management and insecticide-treated net usage. Well-versed communities contribute significantly to active surveillance, promptly reporting cases, and facilitating rapid response efforts critical for elimination. Sustained community participation in long-term elimination endeavours is more likely when communities are knowledgeable about the importance of continued vigilance. A systematic assessment of community knowledge across various countries provides nuanced insights into diverse approaches to malaria programmes. Comparative analysis identifies successful strategies, challenges, and innovations adaptable globally. Outcomes linked to community knowledge help delineate best practices, guiding countries at different stages of malaria control. Examining existing evidence reveals research gaps and areas requiring further exploration, essential for guiding future studies [9].

## Objective

This review is driven by the following objectives:

1. Systematically chart the existing evidence regarding community knowledge of malaria within the context of prevention, control, and elimination programmes executed by various countries.

2. Offer a comprehensive insight into the outcomes linked to the impact of community knowledge in malaria programmes.

Through these objectives, we aim to discover the wealth of knowledge and information surrounding the role of community knowledge in malaria initiatives while facilitating a broader understanding of their effectiveness and implications.

## Materials and methods

This study exclusively focuses on reviewing secondary sources, thereby not requiring approval from a Human Research Ethics Committee. However, this study protocol was registered with

the National Medical Research Register as NMRR ID-23-03703-VZJ. This scoping review will adhere to the Preferred Reporting Items for Systematic Reviews and Meta-Analyses extension for scoping reviews [22]. This framework encompasses a 22-item checklist, which includes two optional elements: a summary of the evidence and a critical appraisal of sources. The checklist will encompass various aspects, including eligibility criteria, search strategy, screening strategy, and data organisation. The comprehensive search strategy will actively encompass an exploration of databases containing peer-reviewed published material, with a particular emphasis on the relevance of community knowledge in malaria prevention, control, and elimination programmes. The search approach is based on the revised guidelines for scoping reviews from the Joanna Briggs Institute [23], which build upon the framework originally established by Arksey and O'Malley [24] and further refined by Levac et al. [25]. Scoping reviews, as an exploratory approach, offer a valid method for synthesising evidence on a specific subject. This approach provides a concise overview of the extent of existing literature and studies without delving into comprehensive analysis [26]. It primarily allows an exploration of various types of evidence, an understanding of the methodologies employed in studies, and aids in the identification and mapping of the evidence within the specified area of interest [26, 27].

## Eligibility criteria

This review will exclusively focus on studies published between the year 2000 to the end of December 2023. This timeframe encapsulates two pivotal milestones—the inception of the Millennium Development Goals (2000–2014) and the ongoing period of the Sustainable Development Goals (2015–2030) [28].

The evidence will be included if it satisfies the following criteria:

- Originating from primary research.

- English language publication

- Employing appropriate methodologies and study designs, including mixed methods designs, qualitative, and quantitative, as well as project reports, case studies, or programme assessments.

- Offering information that directly relates to the role of community knowledge in malaria prevention, control, and elimination.

The evidence falling into the following categories will be excluded:

- Systematic reviews and other reviews.

- Other than English language publication.

- Providing information about the role of community knowledge in contexts unrelated to malaria.

- Texts published as book chapters, abstracts in posters, or conference proceedings.

- No explanation of the study design and methodology in providing evidence.

- Articles involving modelling, simulation, prediction, and machine learning.

## Search strategy

Databases such as PubMed, Embase, Scopus, and Web of Science will be meticulously searched using keywords: "community knowledge" OR "public knowledge" OR "population knowledge" OR "community awareness" OR "public awareness" OR "population awareness" OR "community education" OR "public education" OR "population education" AND "prevention," "control," "elimination" AND "malaria." Keyword selection has been made following the study's objectives. Likewise, we will utilize an advanced Google search to discover grey literature sources, potentially encompassing case reports, program reports, or project reports, employing the same set of keywords. Initially, the search will prioritise titles and abstracts of articles published between 1st January 2000 and 31st December 2023. The comprehensive search will be conducted by FAAR and AMMN across the designated databases. Furthermore, we will strengthen the search by examining the reference lists of the final included articles. In instances where these papers are not available online, the primary author (FAAR) will initiate contact with the corresponding author of the publication via email to request access for review purposes.

## Screening strategy

The screening phase commences following the removal of any duplicate records. It encompasses two stages: title-abstract screening and full-text screening. Before commencing the title-abstract screening, a preliminary batch of 20 articles will be utilised to train the screeners. A concordance rate exceeding 90% between screeners will indicate readiness to initiate the screening process. Each record will undergo independent review by two reviewers for inclusion or exclusion in both the title-abstract and full-text screening phases. Any ties will be resolved by a third reviewer. During the title-abstract screening phase, articles must meet the primary inclusion criteria and at least one of the secondary inclusion criteria. In the full-text screening phase, two reviewers will assess each full text. For inclusion in the full-text analysis phase, all inclusion criteria must be met, and none of the exclusion criteria must be met [29]. In cases where full-text articles are unavailable, the primary author (FAAR) will initiate contact with the corresponding author of the publication via email to request access for review purposes. Following the screening phase, article data will be stored in Microsoft Excel.

## Data extraction process

We will employ a standardised data extraction framework to systematically capture the key attributes of published research literature, ensuring efficiency and consistency throughout the process. This framework will encompass vital bibliographic details, including titles, authors, publication dates/year, study objectives, study locations (countries), study populations, research methodologies, sample sizes, languages, and malaria phases under investigation (prevention, control, elimination). Together, these components will provide a comprehensive summary of crucial study information, facilitating subsequent data analysis. A pilot study will be conducted among reviewers to refine the framework labels, resolving any potential discrepancies or disagreements and enhancing inter-reviewer reliability. Each article will undergo full reading by two reviewers, with conflicts resolved by a third reviewer. Data will then be imported into Microsoft Excel for analysis and summarisation.

## Presentation of the results

We will utilise the PRISMA 2020 flow diagram [30] to visually depict the search process and the progression of evidence across various study stages (S1 Fig). Selected evidence will be

meticulously summarised in a tabular format, presenting source details, study attributes, and key findings. To ensure a comprehensive understanding of the results, we will provide a detailed narrative description aligned with the objectives of the scoping review. Key findings will be synthesised to underscore any limitations and offer an in-depth analysis of the role of community knowledge. This analytical approach will serve as a cornerstone for identifying future research opportunities.

## Discussion

The scoping review conducted in this study spans various timeframes, aiming to establish an initial repository of potentially relevant studies with the primary objective of inclusivity. The title-abstract screening phase is deliberately structured to prevent the inadvertent exclusion of articles containing pertinent information, ensuring the compilation of a comprehensive and inclusive final selection of texts. However, it is crucial to acknowledge the potential limitation of our exclusion criteria, such as book chapters, posters, and conference abstracts, which may result in the omission of valuable data. Additionally, our decision to exclusively consider English-language texts may constrain the review's scope, possibly excluding relevant literature published in languages other than English.

Given that malaria remains endemic in many countries and understanding the role of community knowledge in malaria prevention is still inadequate [31], this scoping study was thoughtfully designed to discover, categorize, and identify research gaps in the context of malaria prevention, control, and elimination. A more structured approach, such as incorporating evidence and gap mapping methodologies, could augment existing evidence identification and strategically identify research gaps. Integrating these frameworks into future reviews can strengthen the synthesis of current knowledge, facilitating a nuanced understanding of areas requiring targeted interventions. The symbiotic relationship between evidence and gap mapping offers a refined roadmap for policymakers, implementers, and community leaders, aiding in developing interventions that harness community knowledge effectively. By integrating these approaches, this review lays the foundation for a more robust and strategic pursuit of malaria elimination goals, ensuring a systematic and comprehensive exploration of the existing evidence landscape.

The results of this scoping review play a pivotal role as they provide a foundational reference point for guiding future efforts and research initiatives in this field. Furthermore, our intention to submit the outcomes of this scoping review to a peer-reviewed journal underscores our commitment to broader dissemination, enhancing accessibility to a wider readership, and invigorating further discourse and exploration in the realm of community knowledge and malaria prevention. This collaborative exchange of insights is essential for advancing our understanding and refining strategies in the ongoing battle against malaria.

## Supporting information

**S1 Checklist. Preferred Reporting Items for Systematic Reviews and Meta-Analyses extension for Scoping Reviews (PRISMA-ScR) checklist [22].**
(PDF)

**S1 Fig. PRISMA 2020 flow diagram for new systematic reviews which included searches of databases and registers only [30].**
(PDF)

## Acknowledgments

The authors thank the Director-General of Health, Malaysia for his permission to publish this scoping review protocol.

## Author Contributions

**Conceptualization:** Faizul Akmal Abdul Rahim, Mohd Hatta Abdul Mutalip, Mohd Amierul Fikri Mahmud.

**Investigation:** Faizul Akmal Abdul Rahim, Mohd Hatta Abdul Mutalip, Ahmad Mohiddin Mohd Ngesom, Mohd Amierul Fikri Mahmud, Norzawati Yoep.

**Methodology:** Faizul Akmal Abdul Rahim, Mohd Hatta Abdul Mutalip.

**Project administration:** Faizul Akmal Abdul Rahim.

**Supervision:** Faizul Akmal Abdul Rahim, Mohd Hatta Abdul Mutalip.

**Writing – original draft:** Faizul Akmal Abdul Rahim, Ahmad Mohiddin Mohd Ngesom.

**Writing – review & editing:** Faizul Akmal Abdul Rahim, Mohd Hatta Abdul Mutalip, Ahmad Mohiddin Mohd Ngesom, Norzawati Yoep.

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
