## [Decision Letter · Decision Letter 0]

2 Feb 2024

PONE-D-24-02013Unveiling the impact of community knowledge in malaria programmes: A scoping review protocolPLOS ONE

Dear Dr. Abdul Rahim,

Thank you for submitting your manuscript to PLOS ONE. After careful consideration, we feel that it has merit but does not fully meet PLOS ONE’s publication criteria as it currently stands. Therefore, we invite you to submit a revised version of the manuscript that addresses the points raised during the review process.

We look forward to receiving your revised manuscript.

Kind regards,

Pisirai Ndarukwa, Ph.D.

Academic Editor

PLOS ONE

Journal Requirements:

2. In this instance it seems there may be acceptable restrictions in place that prevent the public sharing of your minimal data. However, in line with our goal of ensuring long-term data availability to all interested researchers, PLOS’ Data Policy states that authors cannot be the sole named individuals responsible for ensuring data access (http://journals.plos.org/plosone/s/data-availability#loc-acceptable-data-sharing-methods).

Reviewers' comments:

Reviewer's Responses to Questions

**Comments to the Author**

1. Does the manuscript provide a valid rationale for the proposed study, with clearly identified and justified research questions?

Reviewer #1: Partly

2. Is the protocol technically sound and planned in a manner that will lead to a meaningful outcome and allow testing the stated hypotheses?

Reviewer #1: Partly

3. Is the methodology feasible and described in sufficient detail to allow the work to be replicable?

Reviewer #1: No

4. Have the authors described where all data underlying the findings will be made available when the study is complete?

Reviewer #1: Yes

5. Is the manuscript presented in an intelligible fashion and written in standard English?

Reviewer #1: Yes

6. Review Comments to the Author

You may also provide optional suggestions and comments to authors that they might find helpful in planning their study.

Reviewer #1: General comments: The manuscript is well written, however, the current objectives and methods could benefit from further refinement. A general observation was that there was ambiguity in the objectives if they are mapping programs and/or interventions and/ or strategies on knowledge or approaches or programs or evidence. I request the authors to clearly mention what are they seeking specifically (and use the same language throughout the manuscript). Using a variety of terms may create confusion during screening. A suggestion is to select your terms carefully and operationally define it in the methods.

My specific comments on various sections of the manuscript are:

Line 23: Please edit it to "extracting bibliographic details and outcome information" or add more domains to the sentence.

Line 28-29: If the authors mention gaps, they need to strategize how to identify them in this review. Generally (without the presence of framework), a scoping review can find evidence on a particular topic, however, for finding gaps (and evidence both), evidence and gap maps are there.

Line 30: Please edit to "assist" policymakers and empowering others. A suggestion is to mention more implications, currently it is only to design an intervention.

Line 105: Please define, what is community knowledge of malaria

Line 108-109: Avoid using "these" in the objective.

Line 130: Please edit to "Inclusion and exclusion criteria" or "Eligibility criteria"

Line 138-140: The eligibility criteria does not give information about the modelling studies, or studies that may be based on simulation, prediction or machine learning. Please add information in inclusion or exclusion. A suggestion is to not repeat the sentences, eg: including English language articles implies non-English will be excluded.

Line 143: The exclusion can also be according to programs, will you exclude the dengue and chikungunya programs, also give information on what will you do if the study is about all mosquito borne diseases?

Line 144: Please edit, as systematic reviews are not secondary studies, you can write, reviews, eg: systematic reviews and other reviews.

Line 148: Can you specify why multi-country studies will be excluded? It is an important evidence, however, if the authors have a specific reason, please list.

Line 149: Is there any specific reason why book chapters will be excluded, if the book chapter is about a primary study and fits your inclusion, will you exclude it? Abstracts and conference proceedings can be excluded because it is not peer reviewed.

Line 151: Is any search specialist involved in this review, if yes, please describe who will run the search?

Line 153: Please replace "scrutinised" with "searched"

Line 161-162: Please examine reference lists of the final included articles, as the initial search will have many results and to go through each of their bibliographies will be a massive task and delay the review process.

Line 163-164: Please write for how many days will you wait if you do not receive a reply from the authors, a suggestion is to contact the corresponding author as it will be easy to find their email.

Line 165: Please mention if the authors will be using any digital applications (eg: Covidence or Rayyan or EPPI Reviewer) for the process of screening. If the authors will be using Excel spreadsheets, I request the authors to mention that as well. Please write few points about the process of compilation and deduplication in this section.

Line 165: Please mention about the second level screening (full text screening) and how full texts of the articles will be accessed and screened (currently, the authors have only explained title and abstract screening). Please write what will be the procedure the authors will follow if full text (PDF) of the studies are not available.

Line 165: A suggestion is, two authors independently screen each article and a third reviewer can be involved if they authors do not achieve consensus. You can assign one author to review all the excluded title and abstracts (if you want to improve quality further). According to the guidelines if two reviewers achieve consensus, it is adequate for inclusion and exclusion, and involving 4 would not improve the quality of the screening, but you will in turn require double the time. Please consider here, if you can give the required time, you can go ahead with the currently mentioned method. If you have some methods paper which states that quality of screening is better if higher number of reviewers are involved in screening the same set of articles, please provide reference.

Line 172: A suggestion is to pilot the data extraction form with a few articles and edit the form before beginning with the final extraction. This will ensure the authors are able to extract what they intended to.

Line 185: Please use PRISMA 2020 flow diagram (Page et al., 2021) to depict the screening process.

7. PLOS authors have the option to publish the peer review history of their article (what does this mean?). If published, this will include your full peer review and any attached files.

Reviewer #1: **Yes: **Prachi Pundir

---

## [Author Response · Author response to Decision Letter 0]

27 Feb 2024

Dear Dr Prachi Pundir,

We greatly appreciate the feedback you've provided, as it has been instrumental in improving the quality of our article. Rest assured, we are committed to addressing each feedback meticulously and providing reasons if any deviations are made. In this regard, we have included detailed point-by-point feedback to streamline the review process and ensure clarity in our responses.

Thank you.

---

## [Decision Letter · Decision Letter 1]

24 Jun 2024

Unveiling the impact of community knowledge in malaria programmes: A scoping review protocol

PONE-D-24-02013R1

Dear Dr. Abdul Rahim,

We’re pleased to inform you that your manuscript has been judged scientifically suitable for publication and will be formally accepted for publication once it meets all outstanding technical requirements.

Kind regards,

Pisirai Ndarukwa, Ph.D.

Academic Editor

PLOS ONE
---

## [Editor Report · Acceptance letter]

26 Jun 2024

PONE-D-24-02013R1 

PLOS ONE

Dear Dr. Abdul Rahim, 

I'm pleased to inform you that your manuscript has been deemed suitable for publication in PLOS ONE. Congratulations! Your manuscript is now being handed over to our production team.

Kind regards, 

on behalf of

Prof Pisirai Ndarukwa 

Academic Editor

PLOS ONE